# Estimating the Amount of Submerged Marine Debris Based on Fishing Vessels Using Multiple Regression Model

Kyounghwan Song [1], Seunghyun Lee [2,*], Taehwan Joung [3] , Jiwon Yu [2] and Jongkoo Park [4]

1. Air Solution R&D Lab, H&A Company, LG Electronics, Changwon 51554, Republic of Korea; kyounghwan.song@lge.com
2. Maritime Digital Transformation Research Centre, Korea Research Institute of Ships and Ocean Engineering, Daejeon 34103, Republic of Korea; jiwon159@kriso.re.kr
3. International Maritime Research Centre, Korea Research Institute of Ships and Ocean Engineering, Daejeon 34103, Republic of Korea; thjoung@kriso.re.kr
4. A.I. Platform Department, HancomInSpace Co., Ltd., Daejeon 34103, Republic of Korea; jongkoo.park@hancomspace.com
* Correspondence: shlee@kriso.re.kr; Tel.: +82-(0)42-866-3617

**Abstract:** The majority of marine debris is found in shallow waters; however, submerged debris accumulated at the sea bottom is affected by this kind of pollution. To mitigate the harmful effect of marine debris, we have to recognize its characteristics. However, it is hard to estimate the quantity of submerged marine debris because the monitoring of submerged marine debris requires greater cost and time compared to the monitoring of beach or coastal debris. In this study, we used the data for submerged marine debris surveyed in the sea near the Korean Peninsula from 2017 to 2020 and the data of fishing vessels passing through the areas from 2018 to 2020. In addition, the correlation of major factors affecting the amount of submerged marine debris was analyzed based on the fishing vessel data and the removal project data for submerged marine debris. Moreover, we estimated the amount of submerged marine debris based on the fishing vessels at the collection sites surveyed two or more times using a stepwise regression model. The average amount of submerged marine debris estimated by the model was 6.0 tonnes more than that by the removal project, for which the error was ~26.5% compared to the amount collected by the removal project. The estimation method for submerged marine debris developed in this study can provide crucial information for an effective collection project by suggesting areas that require a collection project for submerged marine debris based on the information of fishing vessels.

**Keywords:** fishing vessels; marine debris; sink debris; regression analysis; submerged marine debris

## 1. Introduction

Marine debris is defined as a persistent solid material disposed of in marine and coastal environments. The majority of marine debris is found in shallow waters; however, the sea bottom is affected by this kind of pollution [1]. In addition, submerged marine debris disturbs the oxygen exchange between the upper layer of the sediment and the bottom seawater, damaging the habitat of benthic organisms and affecting the spawning and habitat functions of marine life. In particular, submerged marine debris scattered between sea level and the bottom of the sea damages the habitats of marine life and threatens its growth in Korea [2]. Although studies on submerged debris are less common than those on coastal and beach debris, studies on this topic have increased in recent years [3–11].

When monitoring submerged debris in the sea, researchers need access to marine vessels with the capacity to deploy/retrieve remotely operated underwater vehicles (ROVs) fitted with video cameras [12,13]. Although geographic gaps still exist, some databases, such as the Deep-Sea Debris Database (http://www.godac.jamstec.go.jp/catalog/dsdebris/e/, accessed on 9 January 2022) and LITTERBASE (https://litterbase.awi.de/, accessed on 9

January 2022), provide data on submerged marine debris collected from deep-sea videos and photos taken during research surveys. However, most of the data on submerged debris have been collected in the Northern Hemisphere, and data on deep-sea debris for several regions of the world are spatially and temporarily still scarce [14]. In addition, most of the studies using databases on submerged marine debris in the sea have focused on the pollution assessment of the area where the survey was conducted.

It is hard to estimate the quantity of submerged marine debris because the monitoring of submerged marine debris requires much greater cost and time compared to the monitoring of beach or coastal debris. In this study, we used data on submerged marine debris surveyed in the sea near the Korean Peninsula from 2017 to 2020 and data on fishing vessels passing through the sites from 2018 to 2020. Moreover, the correlation of major variables affecting the amount of submerged marine debris was analyzed based on the fishing vessel data and removal project data of submerged marine debris. We estimated the amount of submerged marine debris based on fishing vessels using a stepwise regression model. The estimation method for submerged marine debris developed in this study can provide crucial information for an effective collection project by suggesting areas that require a collection project for submerged marine debris based on the information from fishing vessels.

## 2. Methods

### 2.1. Data

The data used in this study were the investigated/collected data of submerged marine debris and the data of fishing vessels in Korea. The surveys were conducted by local private organizations and supported by the Ministry of Oceans and Fisheries (MOF) of Korea. The data for the submerged marine debris removal project were collected from 2017 to 2020, which include the date, amount, and location of the investigated/collected debris. Moreover, the data on fishing vessels in Korea, including their trajectories, numbers, tonnage, and license, were collected from 2018 to 2020. Among them, the license information of the fishing vessels was undisclosed data. The Korea Coast Guard (KCG) and the MOF of Korea permitted the use of the fishing vessel data for limited purposes in this study.

Figure 1 shows an example of the collection sites and the trajectories of the vessels passing over the sites, the collection sites that overlapped more than once, and the distribution of the fishing vessels working on the sites (2018–2020) used in this study. Table 1 shows the details of the submerged marine debris removal projects and the number of fishing vessels on the sites selected during 2017~2020 in Korea. A marine debris removal project could include two or more different sites. For example, the number of marine debris removal projects carried out in 2017 was 24, but the number of sites was 35. The collection area means the area calculated without overlap of marine debris removal projects on the sites during the survey period. The number of fishing vessels on the collection site means the total number of fishing vessels on the collection site during the survey period. A fishing vessel was defined as a vessel that passed above the collection site at 5 knots or less. Moreover, the same fishing vessel traveling at less than 5 knots in one day at the collection site was defined as one fishing vessel. The licenses of the fishing vessels were classified into 10 variables by fishing type. Among them, unspecified included those vessels for which no fishing type was defined and/or defined as combined fishing. Additionally, the tonnage of the fishing vessels was classified into 11 variables in units of 1 tonne.

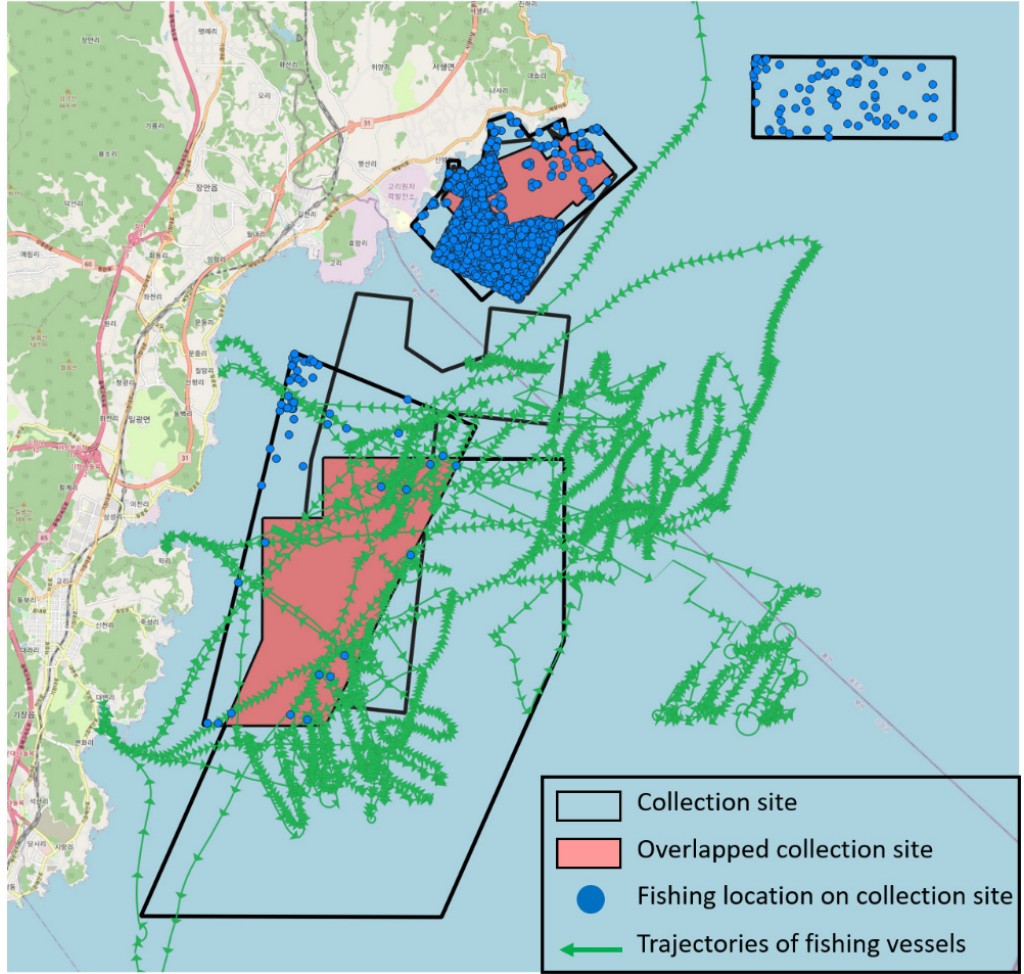

**Figure 1.** An example of the collection sites and the trajectories of the vessels passing over the sites, collection sites overlapping more than once, and the distribution of fishing vessels working on the sites used in this study.

**Table 1.** The details of the submerged marine debris removal projects and the number of fishing vessels on the sites selected during 2017~2020 in Republic of Korea.

| Survey Period (year) | Number of Marine Debris Removal Projects (site) | Amount of Collected Debris (tonnes) | Collection Area (ha) | The Number of Fishing Vessels on Collection Site (count) |
|---|---|---|---|---|
| 2017 | 24 (35) | 1771.55 | 83,018 | - |
| 2018 | 22 (36) | 710.50 | 66,374 | 958 |
| 2019 | 35 (45) | 1484.16 | 72,057 | 11,314 |
| 2020 | 20 (22) | 1053.68 | 60,661 | 124,379 |

*2.2. Statistical Analysis*

All statistical analyses were performed with the free software R 4.1.3 [15]. To evaluate the relationships between the amount of the submerged marine debris and the selected parameters of the fishing vessels, we conducted a correlation analysis. Then, a stepwise regression analysis with both direction was conducted to determine the predictors within the independent variables (i.e., the number of fishing vessels with a license and their

tonnage) which had the most explanatory power over the dependent variable (i.e., the amount of submerged marine debris). Then, 70% of the dataset was used to build the stepwise regression model, and 30% was used for the validation. The equation has the following form:

$$Y = \beta_1(X_1) + \beta_2(X_2) + \cdots + \beta_n(X_n) \tag{1}$$

where Y is the dependent variable; $X_1$, $X_2$, ..., $X_n$ are the independent variables; and $\beta_1$, $\beta_2$..., $\beta_n$ are the standard partial regression coefficients. Equation (1) represents a model of the system under study that can be used to investigate which variables influence its response and to what extent, and/or estimate the value of one variable when the others are known. The best equation was selected based on the highest multiple correlation coefficient. The analyses were carried out in two aspects: for Model 1, the data of the 1st removal project were not considered regardless of whether there were data, and for Model 2, the data of the 1st removal project were considered.

For Model 1, we used the total number of sites where marine debris removal projects were carried out and the number of fishing vessels at those sites shown in Table 2. The accumulation period was not defined. Moreover, the data of the 1st removal project were not considered regardless of whether there were data. Additionally, the number of fishing vessels on the collection site during the period of the 2nd removal project was used. For Model 2, we used data from the sites where the marine debris removal projects overlapped more than two times and the vessels fishing on those sites. It was assumed that all the submerged marine debris on the site where the 1st removal project was carried out was removed. Moreover, the amount of submerged marine debris on the site where the 1st and 2nd removal projects overlapped was calculated as the ratio of the overlapping area to the area of the 2nd removal project. Table 3 shows the number of collection sites and fishing vessels working on the sites during the 1st and 2nd removal projects for Model 2. The amount of accumulated marine debris between the 1st collection project and the 2nd collection project was defined as the amount of submerged marine debris based on the fishing vessels on the site.

**Table 2.** Number of collection sites and fishing vessels working on the sites during the removal project.

| Accumulation Period * (Year) (Not Defined~Removal Project) | Total (Counts) | |
| :---: | :---: | :---: |
| | Collection Sites | Fishing Vessels ** |
| ~2018 | 25 | 958 |
| ~2019 | 32 | 11,314 |
| ~2020 | 22 | 124,379 |

* The accumulation period has been not defined. For Model 1, the data of the overlapped removal project was not considered regardless of whether there were data. ** The number of fishing vessels at the collection site during the submerged debris accumulation period.

**Table 3.** Number of collection sites and fishing vessels working on the sites during the 1st and 2nd removal projects.

| Accumulation Period * (Year) (1st Removal Project to 2nd Removal Project) | Overlapped (Counts) | |
| :---: | :---: | :---: |
| | Collection Sites | Fishing Vessels ** |
| 2017~2018 | 4 | 94 |
| 2018~2019 | 13 | 2269 |
| 2019~2020 | 2 | 6143 |

* If the date of the submerged debris removal was not recorded, the middle date of the removal project was considered as the removal date. ** The number of fishing vessels at the collection site during the submerged debris accumulation period.

### 3. Results

*3.1. Correlation between the Amount of Submerged Marine Debris and Fishing Vessels*

Correlation analysis was conducted to characterize the relationship between the amount of submerged marine debris and the fishing vessels. Then, multiple stepwise regression analysis was used to rank the independent variables that were the strongest predictors of the amount of submerged marine debris. The selected independent predictors were the licenses and tonnage of the fishing vessels. Figure 2 shows the relative importance of the predictor variables for the estimation of the amount of submerged marine debris in Models 1 and 2. For Model 1, the main predictor variables affecting the amount of the submerged marine debris were 0-ton, 3-ton, 4-ton, and 6-ton fishing vessels, and the total number of fishing vessels, as shown in Figure 2a. In particular, 6-ton fishing vessels were the largest contributor to the overall variation in the model, affecting 29.9% of the pattern to estimate the amount of submerged marine debris. Additionally, the fishing vessel licenses affecting the amount of submerged marine debris were anchovy tow nets, purse seines, grill nets, pots, and unspecified, as shown in Figure 2a. The equations of Model 1 are as follows:

$$\text{Amount} = -0.1383(\text{Anchovy tow net}) - 0.4476(\text{Multi-fishing}) + 1.3816(\text{Purse seine}) -$$
$$2.0906(\text{Long line}) - 0.1556(\text{Grill net}) + 6.4046(\text{Trawls}) - 1.6425(0\text{ ton}) + 0.9655(2\text{ ton}) -$$
$$0.8039(4\text{ ton}) + 0.2336(7\text{ ton}) - 1.1761(9\text{ ton}) + 26.0916 \cdot \log(\text{Total vessels}) - \log(\text{Area})$$
$$R^2 = 0.667; F = 5.544; p < 0.00002181$$

$$(2)$$

where the variables are number of fishing vessels by fishing type, number of fishing vessels by tonnage, and area of collection site; $R^2$ is the determination coefficient; $F$ is the fisher ratio; and $p$ is the significance probability level, respectively.

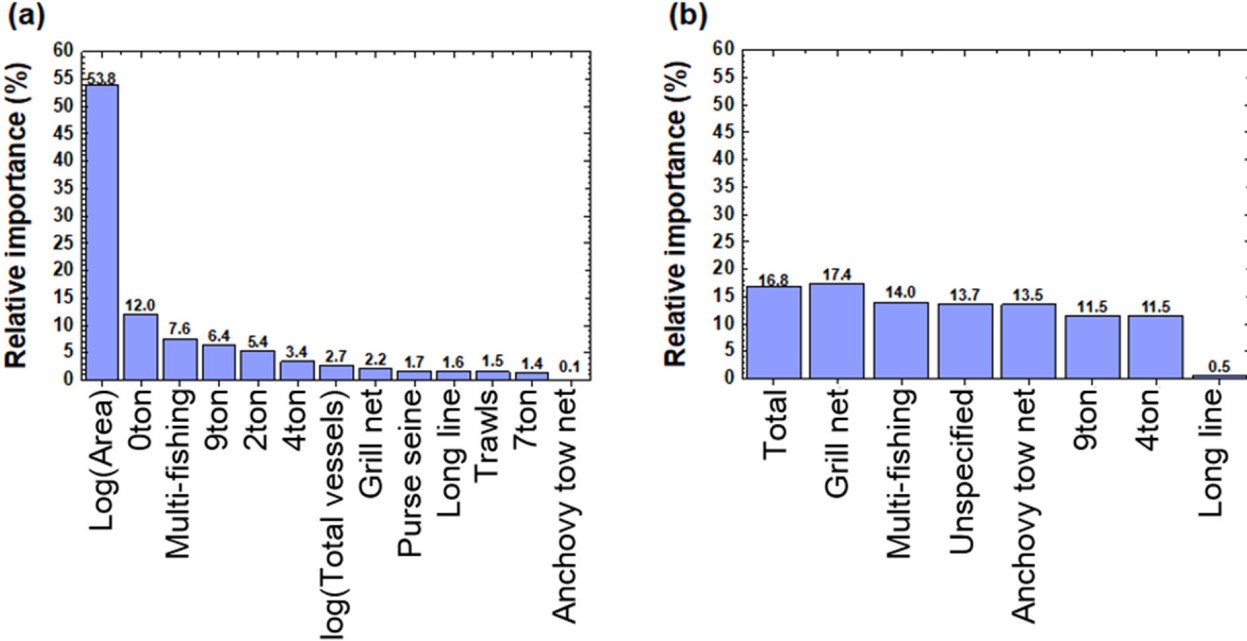

**Figure 2.** The relative importance of the predictor variables for the estimation of the amount of submerged marine debris in (**a**) Model 1 and (**b**) Model 2.

For Model 2, the main predictor variables affecting the amount of submerged marine debris were 4-ton and 9-ton fishing vessels, and the number of total fishing vessels, as shown in Figure 2b. Moreover, the fishing vessel licenses affecting the amount of submerged marine debris were anchovy tow nets, multi-fishing, long lines, grill nets, and unspecified, respectively. Among them, long lines had the lowest relative importance of 0.5%. All the other variables had a similar relative importance from 11.5% to 17.4%. The equations of Model 2 are as follows:

$$\text{Amount} = -0.3771(\text{Multi-fishing}) + 0.2090(\text{Grill net}) - 0.2755(\text{Pots}) + 1.1857(1\text{ ton}) +$$
$$0.4141(2\text{ ton}) + 0.3819(4\text{ ton}) + 16.7435 \cdot \log(\text{Total vessels}) - 3.4292 \cdot \log(\text{Area}) \tag{3}$$
$$R^2 = 0.511; F = 0.785; p < 0.6348$$

where the variables are number of fishing vessels by fishing type, number of fishing vessels by tonnage, and area of collection site; $R^2$ is the determination coefficient; $F$ is the fisher ratio; and $p$ is the significance probability level, respectively.

### 3.2. Model Performance

A quantile–quantile (QQ) plot was used to evaluate the goodness of model fitting. The QQ plot for the standardized residual versus a theoretical normal distribution showed a linear trend. Figure 3 shows the QQ plots of the standardized residuals for Models 1 and 2. Model 1 was closer to linear in the range of the theoretical quantities between −1 and 1; however, some outlier values in the tails of the distribution were also observed shown in Figure 3a.

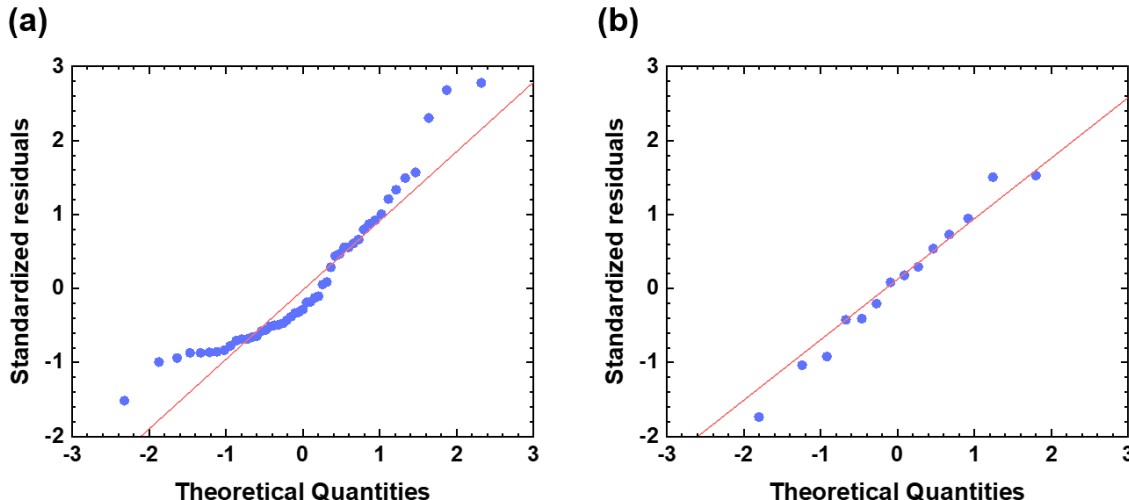

**Figure 3.** Quantile−quantile (QQ) plots of the standardized residuals (**a**) Model 1 and (**b**) Model 2. The lot line and red line represent the data value and the normal distribution, respectively.

### 3.3. Estimation of the Amount of Submerged Marine Debris

The amount of submerged marine debris surveyed by the removal project was compared with that estimated by the two models. Figure 4 shows the amount of submerged marine debris that was estimated by the models developed in this study. It should be noted that the six sites used for the validation were not used to derive the regression model. The amounts of submerged marine debris collected by the removal projects were 30.9, 39.8, 12.3, 25.0, 17.4, and 9.9 tonnes on Sites 1, 2, 3, 4, 5, and 6, respectively. The average amount of submerged marine debris collected by the removal projects on the six sites was 22.5 tonnes. The average amounts of submerged marine debris on the six sites estimated by Models 1 and 2 were 7.1 and 28.0 tonnes, respectively. The average amount of submerged marine debris estimated by Model 1 was 20.9 tonnes less than the average amount collected by the removal project, and the relative error was 68.5%. However, the average amount of submerged marine debris estimated by Model 2 was only 5.5 tonnes greater than that collected by the removal project, for which the relative error was ~24.2% when compared to the average amount collected by the removal project. In particular, the error of Model 2 was the lowest with ~2.1% on Site 4 and highest with 144.6% on Site 3 among the six validation sites.

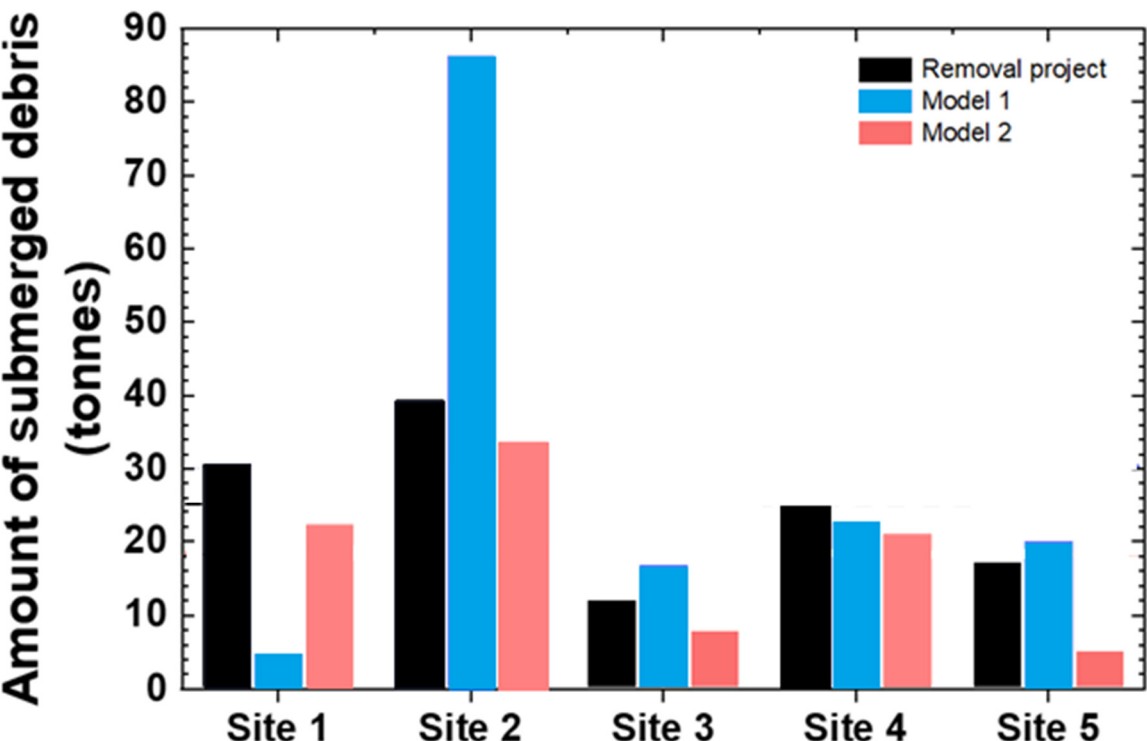

**Figure 4.** Comparison of the models developed in this study. The amount of submerged debris collected by the removal project was considered as the actual standing stock.

## 4. Discussion

### 4.1. Comparison of the Estimation Models for the Amount of Submerged Marine Debris

The most important difference between Models 1 and 2 is that the accumulation period of the submerged marine debris was defined. Sites that overlapped two or more times from different removal projects can define the accumulation period of the submerged marine debris. For Model 1, the period of accumulation of submerged marine debris cannot be defined because the data for the first removal projects were not considered regardless of whether there were data, as mentioned in Section 2.2. However, the data for the fishing vessels used in Model 2 were fishing vessels in overlapping areas of two removal projects during the period between the first and second removal projects. This means that it is possible to clearly define the period of accumulation of submerged marine debris at the overlapping area and the data for the variables that affected the area during that period. Model 2 explains the relationship between the amount of submerged marine debris and the fishing vessels better than Model 1, as shown in Figures 3 and 4. It should be noted that the definition of the accumulation period is a significant factor in estimating the amount of submerged marine debris based on fishing vessels.

### 4.2. Variable Analysis

Equation (3) estimated the amount of submerged marine debris using eight variables, including anchovy tow nets, multi-fishing, long lines, grill nets, unspecified, 4 ton, 9 ton, and total. Table 4 shows the details for the number of fishing vessels with the variables of Equation (3) at the six validation sites. Among the variables, the weight value of anchovy tow nets was the highest at 66.3834, but the number of fishing vessels with anchovy tow nets at all the validation sites was 0. Additionally, the weight value of long lines was the second highest at 30.2889, but the number of fishing vessels with long lines was only 1 on Site 5. Although the weight values of the variables were high, they had little effect on the amount of submerged marine debris at the six validation sites. In addition, it is difficult to clearly explain the relationship between the number of fishing vessels for each major variable and the accuracy of the model. For example, the error of Model 2 was the lowest on

Site 1 with ~26.0%, and that site had the highest number of fishing vessels for the variables among the six validation sites. However, the error of Model 2 was the highest on Site 3 with 118.7, not Site 6, which had the smallest number of fishing vessels with the variable.

**Table 4.** For Model 2, the details of the number of fishing vessels for the variables at the six validation sites, corresponding to Figure 3.

| Variable | Site 1 | Site 2 | Site 3 | Site 4 | Site 5 | Site 6 |
|---|---|---|---|---|---|---|
| Anchovy tow net | 0 | 0 | 0 | 0 | 0 | 0 |
| Multi-fishing | 32 | 11 | 2 | 22 | 1 | 0 |
| Long line | 0 | 0 | 0 | 0 | 1 | 0 |
| Grill net | 111 | 67 | 9 | 15 | 2 | 0 |
| Unspecified | 66 | 27 | 3 | 17 | 6 | 2 |

To explain the amount of submerged marine debris based on the fishing vessels, Model 2, which considers the accumulation period with overlapping sites, was a more suitable model compared to Model 1. However, the major variables selected by the stepwise regression analysis and their weight values need to be corrected. In a stepwise regression model, when two predictor variables are highly correlated with each other, only one variable can exist in the model even if both variables are significant. In addition, the stepwise regression method may incorrectly judge the weights of independent variables to better estimate the amount of submerged marine debris based on the number of fishing vessels. For example, 4 ton, 9 ton, and total in Equation (3) were considered to be variables with a negative weight. This means that they would reduce the amount of submerged marine debris. This could be a fundamental problem of stepwise regression that follows automated rules only considering statistical correlations [16]. However, stepwise regression has become popular in big data because it is a very efficient way of choosing a relatively small number of explanatory variables from a vast array of possibilities. In addition, nine factors selected as predictors in Model 2 show a similar level of relative importance, as shown in Figure 2b. This means that the volatility of Model 2 is not influenced by one particular key factor. Although Model 1 showed higher $R^2$ and F values than Model 2, the accuracy for estimation of the actual submerged marine debris estimate was higher in Model 2, as shown in Figure 4. This means that Model 2 can be improved by more datasets. The amount of datasets is the factor that has the highest influence on the fluctuation of models based on observation data. The developed models will improve the models with continuous data acquisition.

### 4.3. Limitations and Recommendations for Future Research

The total number of collection sites identified by the removal project was 138. However, the number of overlapping sites of the first and second removal projects was 20, as shown in Tables 1 and 3. Although Model 2 estimated the amount of submerged marine debris based on fishing vessels more effectively than Model 1, the dataset was still insufficient. To advance the model developed in this study, it is necessary to carry out additional removal projects on sites where many fishing vessels have been active among the sites where removal projects have been performed in the past.

Investigation of submerged marine debris requires high cost and effort, so a systematic and detailed investigation such as that of coast debris has been lacking. In Korea, most surveys by removal projects have been recording the total amount of submerged debris since the removal project concerns submerged marine debris. However, the properties of submerged marine debris could provide important information for tracing the source of the debris and improving the performance of the model. For example, the amount of submerged marine debris by type (e.g., grill nets, pots, and trawls) could provide important information for evaluating the weight of variables with fishing vessel licenses.

## 5. Conclusions

In this study, we used data on submerged marine debris surveyed in the sea near the Korean Peninsula from 2017 to 2020 and the data of fishing vessels passing over the areas from 2018 to 2020. Additionally, the correlation of major factors affecting the amount of submerged marine debris was analyzed based on the fishing vessel data and removal project data for submerged marine debris. We estimated the amount of submerged marine debris based on fishing vessels operating on the collection sites using the stepwise regression model. The average amount of submerged marine debris estimated by the model considering the accumulation period was 5.5 tonnes more than that estimated by the removal project, in which the error was ~24.2% when compared to the average amount collected by the removal project. The Korean government and the MOF of Korea conduct an annual submerged marine debris removal project. The estimation method of submerged marine debris developed in this study can provide crucial information for an effective collection project by suggesting sites that require a collection project for submerged marine debris based on information for fishing vessels.

**Author Contributions:** Methodology, K.S.; validation, S.L.; investigation, J.P.; writing—review & editing, K.S., T.J. and J.Y.; funding acquisition, S.L. All authors have read and agreed to the published version of the manuscript.

**Funding:** This research was supported by a grant from National R&D Project "Development of Smart Technology to Support the Collection and Management of Marine Debris" funded by the Ministry of Oceans and Fisheries (1525010628).

**Institutional Review Board Statement:** Not applicable.

**Informed Consent Statement:** Not applicable.

**Data Availability Statement:** The original data used to support the findings of this study are partially available through permission from the Korean Ministry of Oceans and Fisheries.

**Acknowledgments:** We sincerely thank the editor and the reviewers for their helpful comments and suggestions about our manuscript.

**Conflicts of Interest:** The authors declare that they have no known competing financial interests or personal relationships that could have influenced the work reported in this paper.

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
