# Peer review of "Estimating the Amount of Submerged Marine Debris Based on Fishing Vessels Using Multiple Regression Model"

_sustainability, doi:10.3390/su152015172_

Round 1

Reviewer 1 Report

Dear Authors

I have thoroughly examined the manuscript and have determined that it possesses commendable quality and offers comprehensive explanations. Nevertheless, it is imperative to provide more clarification for certain assertions:

  • P1, L29-30: choose another word that not provide in title
  • P2, L11-12 and L25-28: The statement of authors is not consistent. at the introduction, it stated the marine debris data and fishing vessel were collected from 2016-2020 and 2017-2020, meanwhile in the method section the data were collected from 2017-2020 and 2018-2020, respectively.
  • P2, L320-48: I am curious in the underlying factor that establishes a correlation between the quantity of submergedmarine debris and the frequency of fishing vessels. Is there any existing literature that suggests a correlation between submerged marine debris and the presence of passing vessels? In addition to the aforementioned point, it is worth noting that the connection between marine debris and passing ships may not be equitable due to the disparity in the temporal scope of the data. Specifically, the garbage data pertains to the year 2016, whilst the ship data corresponds to 2017. Is it feasible for there to be a correlation between two variables if they are measured in the same year?
  • This paper is full of research results but very poor in discussions that can further explain the important things found. For example, what is the main factor influencing the high fluctuation of model 2 based on observation data?  

Best regard

Author Response

First of all, we would like to express our deepest appreciation to the Reviewers for the instructive and kind review of our manuscript, which led us to a much improved presentation of our work. We fully complied with the directions by incorporating the reviewers’ suggestions as far as possible. In the following, we present a list of detailed replies and changes made in accordance with the Reviewers’ comments written in boldface. The changes are highlighted in Italics in the Revised Manuscript.

Reviewer 2 Report

First I want to thank you for the opportunity to read and give constructive comments on this work. This study is relevant to plastic pollution science and global management programs, proposing a novel approach to marine debris surveys.

Title: Estimating the amount of submerged marine debris based on fishing vessels using multiple regression model

Authors: Kyounghwan Song, Seunghyun Lee, Taehwan Joung, Jiwon Yu, Jongkoo Park

General comments:

The manuscript presents an interesting idea, with the ability to be extrapolated to other zones. In a global plastic pollution crisis these approaches could be helpful in management projects around the globe. However, the manuscript is too simple and need more work to exploit all its potential.

I recommended to the authors reorganize the tables in the supplementary material.

The discussion needs more literature for comparison, not only the discussion of their own results but what is going out in Korea. Why did you expect found in waters close to Korea´s sea?

Specific comments:

Methodology:

Page 3; lines 16-30: It is not clear why were generated 2 different models. Please explain better.

Results:

Page 3; Line 40-44: Symbols terminology became a little confusing, please rephrase that to be more clear and easy for the readers. Avoiding the use of a table. Explained better in the method section.

Page 3; lines 45-48 & page 4; lines 8-10: Please reorganize the formula format, it is disorganized.

Discussion:

Page 4; lines 44-46: What did you mean? Please explain better.

Page 5; line 2: Space it needed in  “(…) and Fig. 4 (…)”

Page 5; lines 6-8: Eq. (3) = Model 2? Please be clear and choose one term only.

Author Response

(The authors gave the same response as above.)

Round 2

Reviewer 1 Report

Dear authors

After giving the entire document a thorough read, I can say that all of the issues I raised in the initial round have been resolved satisfactorily. I am pleased to tell you that your manuscript meets all the criteria for acceptance.

All my best